# Biochemical Characterization of the Seed Quality of a Collection of White Lupin Landraces from Southern Italy

**DOI:** 10.3390/plants13060785

**Published:** 2024-03-10

**Authors:** Alfio Spina, Stefano De Benedetti, Giuditta Carlotta Heinzl, Giulia Ceravolo, Chiara Magni, Davide Emide, Giulia Castorina, Gabriella Consonni, Michele Canale, Alessio Scarafoni

**Affiliations:** 1Council for Agricultural Research and Economics (CREA), Centro di Ricerca Cerealicoltura e Colture Industriali, Corso Savoia 190, 95024 Acireale, Italy; alfio.spina@crea.gov.it (A.S.); michele.canale@crea.gov.it (M.C.); 2Department of Food, Environmental and Nutritional Sciences, Università degli Studi di Milano, 20133 Milan, Italy; stefano.debenedetti@unimi.it (S.D.B.); giuditta.heinzl@unimi.it (G.C.H.); giulia.ceravolo@unimi.it (G.C.); chiara.magni@unimi.it (C.M.); davide.emide@unimi.it (D.E.); 3Department of Agricultural and Environmental Sciences, Università degli Studi di Milano, Via G. Celoria 2, 20133 Milan, Italy; giulia.castorina@unimi.it (G.C.); gabriella.consonni@unimi.it (G.C.)

**Keywords:** white lupin, seed composition, legumes, seed proteins, germplasm, anti-tryptic activities

## Abstract

Lupin species provide essential nutrients and bioactive compounds. Within pulses, they have one of the highest contents of proteins and fibers and are among the poorest in carbohydrates. The Mediterranean region is an important cradle area of the origin and domestication of cultivated white lupin (*Lupinus albus* L.). In this work, we present the characterization of 19 white lupin landraces collected from several sites in southern Italy, characterized by different pedoclimatic conditions. The protein contents and electrophoretic patterns, total polyphenols, phytic acid, lipids and phosphorous content, and reducing and anti-tryptic activities have been determined for each landrace. The relationships of the compositional characteristics, the area of origin of landraces and between compositional characteristics and thermo-pluviometric trends that occurred in the genotype comparison field during the two-year period between 2019 and 2020 are compared and discussed. From a nutritional point of view, some of the analyzed landraces differ from the commercial reference. The panel of molecular analyses performed can help in building an identity card for the grain to rapidly identify those varieties with the desired characteristics.

## 1. Introduction

The theme of food security, defined as the economic, social, and physical accessibility to food for all of humanity, is closely related to the issue of environmental sustainability; it is necessary to face such challenges with a view of climate change [1,2] and demographic growth to promote a productive system aimed at exploiting resources in the best possible way while taking care of future generations. The predicted increase in drought for some seasons can affect plant productivity and nutrient availability [3,4,5]. It is therefore important to implement initiatives to enhance new material to be used as such or to be used in genetic improvement programs. Recently, research has been implemented for the identification and characterization of wild varieties or landraces able to grow and develop even in so-called marginal, arid areas that do not need many resources, and at the same time, are sources of high nutritional value [6,7,8,9]. Various wild forms have distinct morphological and nutritional characteristics. In this respect, different aspects are at the basis of the current interest in pulses, such as their peculiar nutritional properties and potential health benefits, the relevance for sustainable agriculture, and the improvement of soil fertility [10].

Among the pulses, lupin provides essential nutrients and bioactive compounds; it has one of the highest content of proteins and fibers and it is among the poorest in carbohydrates [11,12]. The main seed proteins are named α, β, and γ-conglutin [13,14]. The first is the legumin-like globulin and is formed by a family of polypeptides ranging from about 37 kDa to 54 kDa, all linked to a smaller polypeptide of 21 kDa. β-conglutin is the vicilin-like globulin that quantitatively represents the major lupin seed proteins [15], and whose principal polypeptides fall in the 44–64 kDa regions; no disulfide bonds are present in the protein. Its roles in plant defense and human health benefits have been described [16,17]. γ-conglutin is made by six monomers of about 46 kDa, made up of two subunits of 17 and 30 kDa linked by a single disulfide bond [15]. This protein has been studied for its postprandial glycemic regulating activity [18]. The interesting nutritional characteristics and nutraceutical properties [17,18,19,20,21] make the lupin attractive to consumers and offer new opportunities for the cultivation of lupin worldwide. In particular, the Mediterranean region is an important geographical cradle area of the origin and domestication of cultivated white lupin (*Lupinus albus*). The germplasm from this region shows a great diversity of agronomic, morphological, physiological, and qualitative traits [22].

Landraces are useful sources of variants in genes controlling the traits of agronomic importance, such as disease and abiotic stress resistance, resilience to climate changes, and seed quality [23]. They may be exploited for the introduction of desired characteristics into modern cultivars. The processes of domestication and breeding of lupin mainly focus on improving yield as well as consumer and animal acceptance by reducing the content of quinolizidine alkaloids, but more efforts are needed to face the current challenges. In this context, we have been collecting *Lupinus albus*, *L. luteus,* and *L. angustifolius* samples for a decade [24] from several sites in the southern regions of Italy, all characterized by different pedoclimatic and growing conditions. We conducted a first screening for the content and composition of quinolizidine alkaloids on eighteen genotypes of *Lupinus albus*, three of *L. luteus* and one of *L. angustifolius*, and found that all landraces, unlike the improved varieties, had a very high content of quinolizidine alkaloids [24].

The present work is a continuation of this line of research and aims to depict a complete characterization of seeds from 19 *L. albus* landraces. Two main approaches have been taken into consideration: the first includes the macromolecular composition, while the second consists in the determination of molecular features related to the bioactivities relevant to human wellbeing.

## 2. Results and Discussion

### 2.1. Samples Description

This work illustrates the biochemical composition of 19 lupin landraces and one commercial breeding (*Lupinus albus*, Ares), with a focus on those parameters that are relevant to human and animal nutrition. In Table 1 are the reports of all the analyzed lupin landraces and their specific region/province of collection. They originate from different regions of southern Italy (Figure 1) and were grown in a single site following the same agronomic technique to minimize the possible differences due to annual pedoclimatic conditions.

### 2.2. Description of the Main Climatic Data

The rainfall trend recorded over the 2019–2022 two-year period was characterized by average rainfall but poorly distributed throughout the biological crop cycle. In particular, the second ten days of December to the second ten days of March were characterized by an abnormally low rainfall. In contrast, abundant and exceptional rainfall during the third ten days of March ensured regular pods and seed growth and development.

Considerable temperature rises were recorded from the second ten days of April, with particularly high values in the second ten days of May. The thermal trend in the third ten days of June (harvest time) was more regular, with temperatures in the normal range and maximum values close to 30 °C. The rainfall trend for the 2019–2020 two-year period is consistent with the ten-year (2010 to 2019) average rainfall. Similarly, the minimum, maximum, and average temperature values of the ten years are also in line with those recorded in the 2019–2020 two-year period (Figure 2).

Therefore, all qualitative data obtained in this study may be considered consistent with the characteristics of each genotype, as each was able to express its biochemical peculiarities.

### 2.3. Plants Phenological Features

During the sowing period, a minimum temperature of 9 °C, a maximum of 21.4 °C, and an average temperature of 15.2 °C were recorded, but there was no rainfall. After 3 days, light rainfall (almost 3 mm) occurred, which promoted seed swelling and activated germination enzymes. The emergence of the different white lupin genotypes was recorded within a short time interval (from 11 days in Leonforte 5 and Leonforte 6, Canicattini and Grammichele to 15 days in Calabria 3 and Molise). Thus, the data indicate that during the 5-day time frame, the emergence of all evaluated ecotypes was recorded.

There was a significant statistical difference (*p* < 0.001) between the observed samples in terms of main stem flowering emission and a high correlation coefficient (r = 0.89) compared to the end of germination time. This variability is due to earlier ecotypes such as Grammichele, Canicattini, and Leonforte 6, which started flowering 78, 80, and 81 days after sowing, respectively. Other ecotypes, like Calabria 3 and Molise (112 and 109 days after sowing, respectively), flowered later.

All other genotypes flowered in a time interval between these two groups of landraces.

This observed early germination influenced pod development, confirming a significant statistical difference (*p* < 0.001) between the samples. The first ecotypes to produce and develop pods were Grammichele (91 days after sowing), followed by Canicattini, Leonforte 6, and Acireale at 95, 96, and 98 days after sowing, respectively. All the other genotypes formed pods at 103 days after sowing, with a high correlation coefficient (r = 0.87) to sowing time. Molise, Puglia, Lecce, and Calabria 3 were the last.

Regarding pod and seed ripening, most genotypes showed an early seed of maturity (around 175 days after sowing). On the other hand, the ecotypes Calabria 3, Molise, and Puglia had the longest biological cycle and reached maturity later, with a high correlation coefficient (r = 0.92) to the sowing date (Table 2).

### 2.4. Characterization of Biochemical and Nutritional Features

For quantitative characterization, proteins were extracted from the milled flour of each seed sample and quantified through the Bradford assay. The results are reported as grams of protein per kg of dry weight (g/kg DW) of milled seeds (Appendix A). The total extractable protein content was variable and spanned from the 223 ± 2 g/kg DW of Calabria 2 landrace (namely 22.3%) to the 468 ± 33 g/kg DW detected in the Basilicata ecotype (46.8%). Overall, the data are in good accordance with the mean protein content already published for white lupin, which is around 35% [25]. Commercial reference Ares presented a high protein content (394 ± 5 g/kg DW), not statistically different from the landraces with the highest protein quantification; landraces from Sicily presented a lower variability in their protein content, while a higher variability was instead observed among landraces from the Calabria region. Besides protein content, several other qualities relevant for the nutritional characterization of lupin seeds were assessed.

Lipid content was low, as expected, with values between 51 ± 2 g/kg DW (Scicli) and 108 ± 4 g/kg DW (Leonforte 2). Most of the analyzed samples, in particular those originating from Sicily, with the mentioned exception of Scicli landrace, showed little variability in lipid content, settling at values around 90 g/kg DW (namely 9%), a value slightly lower than what can be found in the literature [26,27,28,29]. Because of this low lipid content, lupin is not interesting for oil extraction, unlike soybean whose oil content is up to 24% [30]. However, it is valuable from a nutritional point of view, for the quality of its fatty acids [31].

Polyphenols represent another interesting class of molecules for nutrition, due to their ability to contrast oxidation. This is relevant also from a food technology point of view; the presence of molecules with a reducing power may help to increase the shelf life of food products through enhancing lipids stability, thus maintaining nutritional and sensorial properties unaltered [32]. The sample with the highest polyphenol content was Canicattini (10.4 ± 0.6 mg Quercetin Equivalent/g). Other lupin landraces showed overall comparable values. Similar polyphenol contents were observed in different genotypes of *Lupinus* species from Andalusia, Spain, where the content ranged between 8.7 mg/g flour in *L. micranthus* and 11.0 mg/g flour in *L. angustifolius* [33]. In addition, Krol et al. [34] detected a polyphenol content in from the dry weight of narrow-leaf lupin seeds ranging from 1.89 to 8.1 mg/g, with the highest polyphenol content in seeds rich in alkaloids. Siger et al. [35] analyzed phenolic compounds present in extracts from seeds of *L. albus*, *L. luteus,* and *L. angustifolius* cultivars. According to these authors, total phenolic content varied from 491.51 to 731.14 mg/100 g of dry matter. The landraces selected for this study, thus, have a polyphenol content similar to what is reported in the literature. Finally, reducing power was evaluated with the FRAP method. Overall, the analyzed landraces showed little difference in reducing power attesting around an average of 130 mmol equivalent FeSO_4_/g. However, Acireale landrace stands out, showing the highest reducing power (235.7 ± 11.7 mmol Eq FeSO_4_/g).

Phosphorous content was evaluated as closely related to phytic acid content, a known antinutritional component of plant-based foods. The phosphorous content is constant over all the analyzed landraces, as well as the phytic acid content. The latter is, indeed, very low in all the analyzed samples, between 2.1 ± 0.1 and 3.0 ± 0.1 g/kg DW (0.21–0.30%), if considering that in legumes, phytic acid content generally ranges between from 0.40% to 2.06% [36] and values of 2.32% are reported for white lupin [37]. Other relevant antinutritional factors in legumes are trypsin inhibitors because they can reduce digestion and the absorption of dietary proteins [38]. Inhibitory activity was then evaluated and interestingly, it results were widely variable: several landraces (Leonforte 2, Leonforte 4, Calabria 3, Puglia, Basilicata, Molise) showed no inhibitory activity at all (negative values for TIA reported in Appendix A were considered as zero inhibitory activity, due to experimental uncertainty), thus gaining particular attention for nutritional purposes.

To facilitate the visualization of all the analyzed variables, a graphical representation through radar plots was chosen (Figure 3, Figure 4, Figure 5 and Figure 6). In this way, each analyzed lupin landrace is described by a characteristic fingerprint that recaps all the determinations performed and enhances the differences. This representation helps to compare such different and numerous variables and to rapidly identify the similarities, differences, and uniqueness of the analyzed landraces. According to this comparison, it is possible to highlight that the commercial reference Ares is the variety with the highest trypsin inhibitory activity (TIA) and the lowest polyphenol content, attributing to this variety of undesirable characteristics.

All Leonforte landraces have similar protein, lipid, and polyphenol content. Only Leonforte 3 and 4’s protein content is significantly lower than Ares, but all Leonforte landraces have a significantly higher polyphenol content than the reference. However, they widely vary for TIA, with Leonforte 5 among the landraces with the highest activity. Grammichele presents traits similar to Leonforte samples, indeed both originating from Sicily but from different areas. Landraces originating from southern Sicily present more varied plot shapes, if compared to Leonforte. Overall, they have a higher protein content than Leonforte, but only Scicli is significantly higher. In landraces from southern Sicily, lipid content is more variable than in those from central Sicily, with Scicli presenting the lowest lipid content. Also, polyphenol content and FRAP activity is more variable among these samples; in this group are present those landraces with the highest values: Canicattini for polyphenol content and Acireale for reducing power. Except for Calabria 2, landraces from the Calabria region have a comparable protein content with Ares. The Puglia and Molise landraces have similarities, since showing comparable contents of all the evaluated compounds. Calabria 3 and Basilicata also showed a similar shape of the radar plot, particularly consequent to the high protein content and low TIA. The landrace with the most interesting combination of positive nutritional factors is Canicattini, where a good protein content is associated with high polyphenol amounts and a low content of antinutritional factors can be observed. Among those landraces with low or no trypsin inhibitory activity at all, Calabria 3 and Basilicata are interesting due to their high protein content; however, they both have a low polyphenol content. All of the analyzed landraces present high variability in their composition, which is likely dependent on their intrinsic genetic variability. Further analyses will be performed to explore this variability. Nevertheless, the kind of compared analysis performed in this work could be proposed as a helpful approach in rapidly identifying landraces with the desired combination of characteristics.

When considering so many different variables, multivariate statistical analysis may help to cluster different observations and disclose hidden relationships between variables. For this purpose, a Principal Component Analysis (PCA) was performed (Figure 7). The goodness of the analysis is confirmed by the evaluation of the correlation matrix, which highlights a strong positive correlation between phosphorous and phytic acid content, and a positive correlation between polyphenol content and reducing power (Appendix A). It is interesting to highlight a positive correlation as well between TIA and phytic acid content that suggests a relationship between antinutritional factors. Interestingly, PCA highlights that the Ares reference sample is well discriminated from the landraces analyzed, indicating that the work presented describes a set of lupin landraces with characteristics different from what is already present in the market. Strikingly, Ares is characterized by those variables that should be undesirable in a food product: high lipid and antinutritional factors content, and low polyphenols. Most of the analyzed landraces, independently from the area of origin, are clustered in the left panel of the Biplot (Figure 6), characterized by those variables that could be referred to as beneficial from a nutritional point of view, such as high proteins and polyphenols and low lipids and antinutritional factors. Finally, the PCA clusters Leonforte 5 and Leonforte 6 as varieties with a high content of antinutritional factors.

The attribution of peculiar nutrient content profiles to each landrace may help to select the appropriate one as a function of the desired application. Lupin indeed, due to its high-protein content, is a good gluten-free meat substitute in vegetarian and vegan diets; furthermore, due to its low lipid content. it could be even better than soybean who, nowadays, is the gold standard. From a technological point of view instead, a high content of polyphenols and/or molecules with a reducing power could help in extending shelf life or enhance food properties in particular products, adding to the other well-known emulsifying, foaming, and gelling properties of lupin seeds [39].

### 2.5. Qualitative Protein Characterization

Proteins being the main fraction of lupin seeds, we also evaluated the qualitative differences in the protein pattern through SDS-PAGE analysis (Figure 8). A reducing condition was chosen to make a clear distinction between the main seed proteins, namely α-, β- and γ-conglutin [33,34]. The first is the legumin-like globulin, which is formed by a family of polypeptides ranging from about 37 kDa to 54 kDa, all linked to a smaller polypeptide of 21 kDa. β-conglutin is the vicilin-like globulin that quantitatively represents the major lupin seed proteins [35], and whose principal polypeptides fall in the 44–64 kDa regions; no disulfide bonds are present in the protein. The roles in plant defense and human health benefits have been described [17,36]. γ-conglutin is made of six monomers of about 46 kDa, made up of two subunits of 17 and 30 kDa linked by a single disulfide bond [35]. This protein has been studied for its postprandial glycemic regulating activity [14]. To perform a qualitative analysis of the protein pattern, a sample subset was selected that included the landraces with the highest (Basilicata, Calabria 1, Calabria 3, Calabria 4, Modica, Canicattini, Scicli) and the lowest (Calabria 2, Molise, Puglia) total protein content. In addition, a commercial variety (Ares) and Grammichele, as an intermediate protein content landrace, were also included. A reducing condition was chosen for the SDS-PAGE analysis to make a clear distinction between the different protein families.

The band corresponding to the main seed proteins [13,25], namely α-, β-, and γ-conglutin, are evident in all the analyzed samples, regardless of their total protein content, as indicated by arrowheads in Figure 7. Overall, the electrophoretic approach showed that the protein profile of the analyzed landraces was very similar to each other and as well to the commercial reference Ares. However, some differences can be highlighted in the intensity of the specific band on the gel. Compared to the control, the landraces have a lower relative content in γ-conglutin, especially Calabria 1 and Calabria 2. This result highlights that commercial Ares is slightly richer in hypoglycemic protein. Instead, the amount of β-conglutin, the protein involved in plant defense, is similar in all samples analyzed.

## 3. Materials and Methods

### 3.1. Plant Material and Sampling

The seeds of 19 Italian landraces of *Lupinus albus* were collected from different southern Italian regions. The names of the Italian landraces are the same as the city or region from where they were harvested. Leonforte 1–6 originate from the central region of Sicily; Acireale, Canicattini, Grammichele, Modica and Scicli originate from the south-east region of Sicily; the samples named Calabria 1–4 originate from the Calabria region; the samples Lecce and Puglia are landraces originating from the Apulia region, while landraces Molise and Basilicata originate, respectively, from the Molise and Basilicata regions. To minimize the possible differences due to annual pedoclimatic conditions, all plants were grown in a single site following the same agronomic technique.

Seeds of all studied genotypes were deposited within the germplasm collection of the Council for Agricultural Research and Economics (CREA)—Research Centre for Cereal and Industrial Crops, Laboratory of Acireale (Acireale, Italy). Furthermore, a commercial lupin breeding (*Lupinus albus* Ares) was used as reference material. The seeds were dried at 50 °C in an oven until no further weight loss was recorded (48 h), finely ground to 60 mesh particles, and stored in sealed jars.

### 3.2. Experimental Site and Climatic Details

The trial was conducted in 2019–2020 on volcanic and acid soil in East Sicily, Giarre, Italy. All seed samples were sowed in duplicated plots of 5 m^2^ (2.5 m× 2 m). Manual seeding was completed on 16 December 2019. Fertilization was applied during sowing with 30 kg ha^−1^ of ammoniacal nitrogen (ammonium sulfate 20–21 N) and 60 kg ha^−1^ of phosphorous (mineral perphosphate 18–20 P_2_O_5_). Weed control was carried out twice mechanically post emergence (at the fifth true leaf and just before flowering). Flood irrigation was required from the end of March to mid-June. Aphicide treatment with 50 mL/hL of Imidacloprid (Confidor, Bayer CropScience, Milan, Italy) was applied in late March. The crop was harvested on 23 June 2020. A field trial with yellow, narrow-leaved and white lupin genotypes (including the Modica ecotype) was conducted nearby (Acireale) in 2013/2014 following similar agronomic management [40].

The experimental environment is typical of the coastal area of eastern Sicily (Cfa climate, according to Köppen and Geiger [41]), with a long, dry summer period and a colder winter, with no snow days and irregular rainfall. The data are reported in Figure 8. The average annual rainfall is about 850–950 mm, mostly in the autumn–winter period, while in the spring it amounts to about 25–30% of the annual rainfall. The summer months are mostly dry, with torrential rainfall. In recent years there has been a decrease in autumn rainfall and an increase during the spring.

The total rainfall during the lupin crop cycle (Figure 2) reached 399.4 mm, of which 182 mm occurred in the third ten-days of March.

Temperatures were high, as usual in the experimental area, with average values of 13.7 °C and maximum values between 22.3 and 33.5 °C at the end of the crop cycle.

December and February were the driest months. March was the wettest month with 189.6 mm of rain. On the other hand, rainfall was lower (51 mm from the second ten days of December, i.e., from sowing, to the second ten days of March) (Figure 8). The lowest temperatures were recorded during the first ten days of January (4.1 °C), the second ten days of February (3.1 °C) and the third ten days of March (3.8 °C).

### 3.3. Plants Phenological Measurements

During the entire lupin biological cycle, the time in which the main phenological phases occurred was noted, and measurements were carried out as the days elapsed from sowing to the various phenological phases, such as end germination–beginning plant emergence, main stem flowering, main pods growth, and pod and seed maturity.

### 3.4. Protein Extraction and Quantification

One gram of each flour was resuspended in 20 mL (ratio 1:20 *w*/*v*) of 50 mM Tris-HCl buffer, pH 9.0, containing 300 mM of NaCl. After stirring for 3 h at 4 °C, the suspensions were centrifuged at 12,000 rpm for 45 min at 4 °C. The supernatant was transferred into Eppendorf tubes and centrifuged again for 25 min at 4 °C. The supernatant was collected for further analysis. The protein extracts were diluted (1:100) in a phosphate buffer before Bradford assay [42]. Then, 1 mL of Bradford reagent was added to 100 µL of each sample and absorbances were read at 595 nm with a spectrophotometer (Lambda 2, Perkin-Elmer, Waltham, MA, USA). Sample blanks were also prepared with 100 μL of dH_2_O and 1 mL of Bradford reagent. The protein concentration was plotted against the standard calibration curve. Measures were performed in duplicate.

### 3.5. SDS-PAGE

Protein extracts were analyzed by SDS-PAGE, separated on 12% polyacrylamide gel with 4.5% stacking gel, according to Laemmli [43], mixing supernatants with the same amount of loading buffer solution in the presence of β-mercaptoethanol and boiled for 10 min to allow the denaturation to occur. An amount of 15 µL per sample were loaded on the gel. The molecular mass markers were b-phosphorilase (92 kDa), BSA (66 kDa), egg albumin (45 kDa), carbonic anhydrase (30 kDa), trypsin inhibitor (20 kDa), and lysozyme (14 kDa). The run was carried out at 16 mA, constant for each gel for 90 min on a Mini-Protean III Cell (Bio-Rad Laboratories, Inc., Hercules, CA, USA); the gels were stained with Coomassie Blue G-250 (BioRad, Milan, Italy) and destained with a 10% ethanol, 10% acetic acid solution. Images were elaborated with Image Lab™ (version 6.0.1, Bio-Rad Laboratories, Inc., Hercules, CA, USA).

### 3.6. Lipids Quantification

Lipid extraction was performed in agitation from 0.3 g of flour in 1.5 mL of cold pentane for 1 h. After the extraction, the samples were centrifuged at 12,000 rpm for 5 min. The supernatant was collected in Eppendorf and the solvent evaporated under a fume hood. The extraction was performed four times, and the supernatants from each sample were collected in the same tube. To eliminate the residual solvent, the tubes were placed at 50 °C overnight. The results were calculated as follows:$$Lipids\;\%=\frac{me}{ms}*100$$
where:

*ms* = mass of flour (g)

*me* = mass of extracted oil (g)

### 3.7. Trypsin Inhibitor Activity

Trypsin inhibitory activity (TIA) was measured according to ISO 14902 standard method [44], with slight modifications. Trypsin activity was quantitatively determined using the synthetic substrate *N*-Benzoyl-l-arginine 4-nitroanilide hydrochloride (BAPA, Merck Life Science, Milan, Italy). Trypsin stock solution was prepared by dissolving 2.7 mg of trypsin (Merck Life Science, Milan, Italy) in 10 mL of 1 mM HCl with 5 mM CaCl_2_. This solution was then diluted 1:10 prior to the test execution. A BAPA working solution was obtained by diluting 1:100 of the stock solution (1.5 mM in DMSO) in 50 mM of Tris-HCl, pH 8.2, and 5 mM CaCl_2_. The assay was performed by mixing 0.1 mL of protein extract with 0.8 mL of BAPA working solution and 0.1 mL of trypsin solution. After 10 min of incubation at 37 °C, the absorbance was read at 410 nm. A reference value was realized using dH_2_O instead of the sample solution. Sample blank and reference blank were realized in the absence of the inhibitor by replacing it with an equal amount of dH_2_O.

TIA was calculated as:$$Inhibition\%=\frac{\left(Ar-Abr\right)-\left(As-Abs\right)}{Ar-Abr}*100$$

*Ar* = absorbance of reference

*Abr* = absorbance of reference blank

*As* = absorbance of sample inhibitor

*Abs* = absorbance of sample blank

Each experiment was performed in triplicate.

### 3.8. Reducing Activity

Reducing activities were analyzed through the Ferric Reducing Ability of Plasma (FRAP), developed by Benzie and Strain in 1996 [45]. For the extraction, 10 mL of ethanol (80% *v*/*v*) were added to 0.5 g of flour and stirred for 24 h at 22 °C. Then, the suspension was centrifuged at 2370× *g* for 30 min at 4 °C. The supernatant was recovered, filled to 10 mL of volume with extraction solvent, and kept at −20 °C until used. Then, the assay was carried out with 3 mL of working FRAP solution (25 mL acetate buffer 300 mM, pH 3 added to 2.5 mL 2,4,6-tipyridyl-s-triazine (TPTZ) (Merck Life Science, Milan, Italy) 10 mM in HCl 40 mM and 2.5 mL FeCl_3_ 20 mM in dH_2_O), and 400 µL of samples. The reaction was stopped after 3 min and absorbances were read at 593 nm. Each experiment was performed in triplicate. A sample blank was realized with 400 µL of solvent used for extraction. A calibration curve was prepared with FeSO_4_·7H_2_O between 200 and 1000 µmol/L. The results were expressed as millimoles of FeSO_4_·7H_2_O equivalents per gram of flour.

### 3.9. Phytic Acid and Phosphorus Content Assay

Phytic acid and phosphorous content were analyzed with a Megazyme K-PHYT kit (Megazyme, Wicklow, Ireland), according to manufacturer instructions. The total phosphate released was measured by the increase in absorbance at 655 nm and given as grams of phosphorus per 100 g of sample material.

### 3.10. Total Phenolic Extraction and Quantification

The total phenolic content (TPC) was measured by applying the Folin–Ciocalteu reagent method [46] with some modifications. A total of 250 mg of sample was added to 5 mL of ethanol and 50 mL HCl at 30 °C overnight, for phenolic compound extraction. After the suspension was centrifuged at 2000 rpm for 30 min, the supernatant was recovered and kept at −20 °C until used. For phenolic quantification, 50 µL of extracts were added to 0.5 mL of Folin−Ciocalteu’s reagent, 2 mL of 2 M Na_2_CO_3_, and dH_2_O up to 10 mL final volume. After incubation for one hour at room temperature, the absorbance was measured at 765 nm using a spectrophotometer (Lambda 2, Perkin-Elmer, Waltham, MA, USA). A calibration curve was prepared with serial dilutions of a quercetin standard (Sigma-Aldrich, St. Louis, MO, USA) concentrated 0.5 mg/L, in a range comprised from 0.5 mg/L to 0.025 mg/L. The linear equation obtained from the standard curve was used to interpolate each sample of total phenolic extract concentration, and the results were reported as milligrams of quercetin equivalents (QE) per gram of flour.

### 3.11. Statistical Analysis

Statistical analysis and graphical reports have been performed with Origin Pro 2023 (OriginPro, Version 2023. OriginLab Corporation, Northampton, MA, USA).

The results are reported as mean ± standard deviation. Mean comparisons were performed with Tukey’s Test. The significance level was set at 0.05 and it is reported in Appendix A with the lowercase letters in the brackets; the same letter means the difference is not statistically significant.

The correlation coefficient (r) between the sowing time and the four phenological stages has been calculated.

## 4. Conclusions

Lupin seeds are a good source of essential nutrients and bioactive compounds. Thanks to its peculiar composition, this legume represents an interesting alternative to foods of animal origin, also considering its added value of having little environmental impact. The molecular characterization of the grain quality opens the way to drive personalized uses both as a food and as a techno-functional ingredient, addressing the choice of the most appropriate variety, for the most appropriate destination.

In this work, 19 *L. albus* landraces were characterized to identify similarities, differences, and uniqueness.

The thermo-pluviometric trend for the 2019–2020 two-year period is consistent with the ten-year mean (2010 to 2019). Therefore, all qualitative data obtained in this study are consistent with the characteristics of each genotype, as each was able to express its biochemical peculiarities.

Overall, the results revealed a wide variability among the analyzed landraces and also several differences in the biochemical composition with the commercial reference. In particular, the commercial Ares cultivar presents a good protein content, but it is the variety with the highest trypsin inhibitory activity (TIA) and the lowest polyphenol content, which are undesirable characteristics.

The comparative analysis performed using radar plots showed that most of the landrace analyzed presents an interesting combination of positive nutritional factors. Indeed, high protein and polyphenols content is associated with low content of antinutritional factors.

The molecular characterization described in this work can act as an identity card for the grain and represents a method useful for the identification of varieties with desired characteristics for a specific destination. The outcomes provide scientific information and molecular tools to develop new varieties and thus boost the practical use of white lupin as a food and feed source in sustainable agriculture.

## Figures and Tables

**Figure 1 plants-13-00785-f001:**
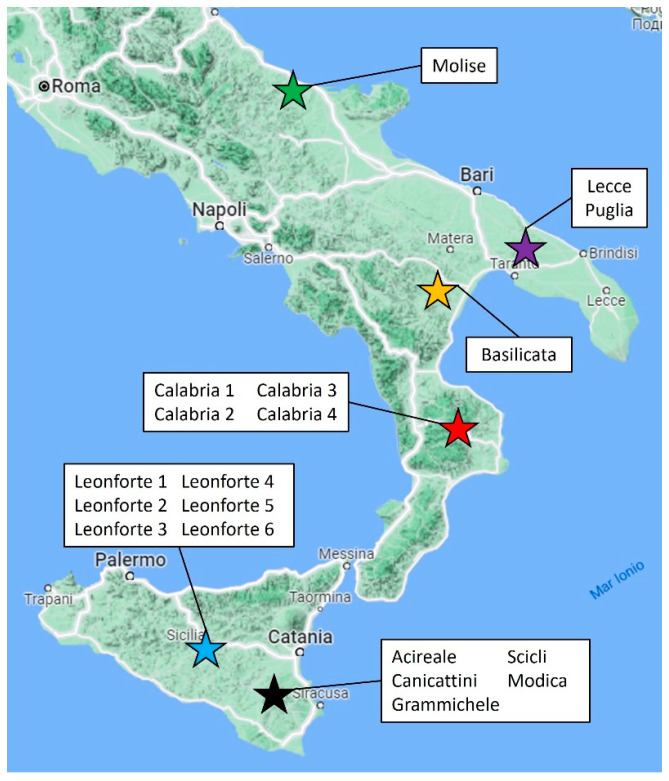
Southern Italy: areas of origin of white lupin landraces.

**Figure 2 plants-13-00785-f002:**
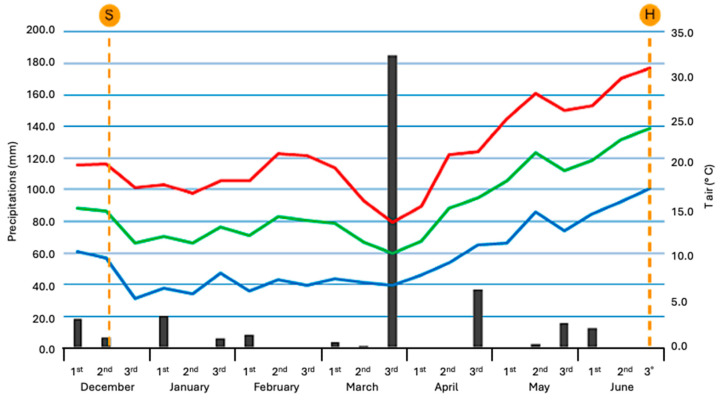
Cumulated rainfall (mm; black bars) and maximum (°C; red line), minimum (°C; blue line), and mean (°C; green line) temperatures recorded in 2019–2020 at Giarre, Catania, Italy. Data are reported as ten-day values from December to June. S: Sowing time; H: Harvest time.

**Figure 3 plants-13-00785-f003:**
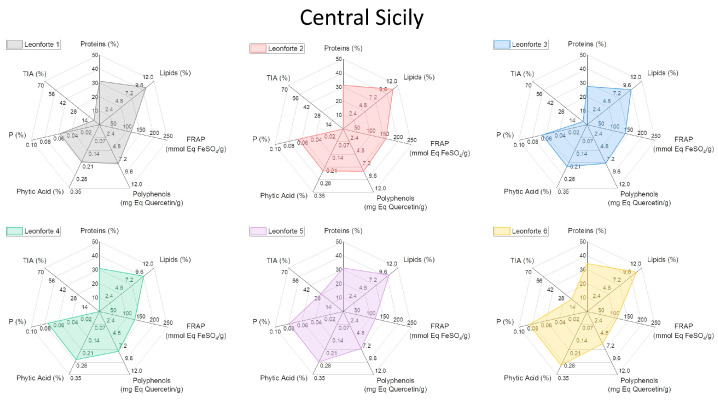
Radar plot representation of the main biochemical characteristics of white lupin landraces collected in central Sicily. See Appendix A for detailed information.

**Figure 4 plants-13-00785-f004:**
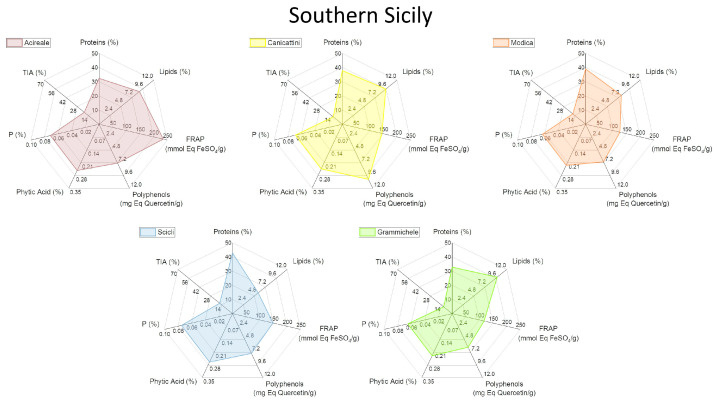
Radar plot representation of the main biochemical characteristics of white lupin landraces collected in southern Sicily. See Appendix A for detailed information.

**Figure 5 plants-13-00785-f005:**
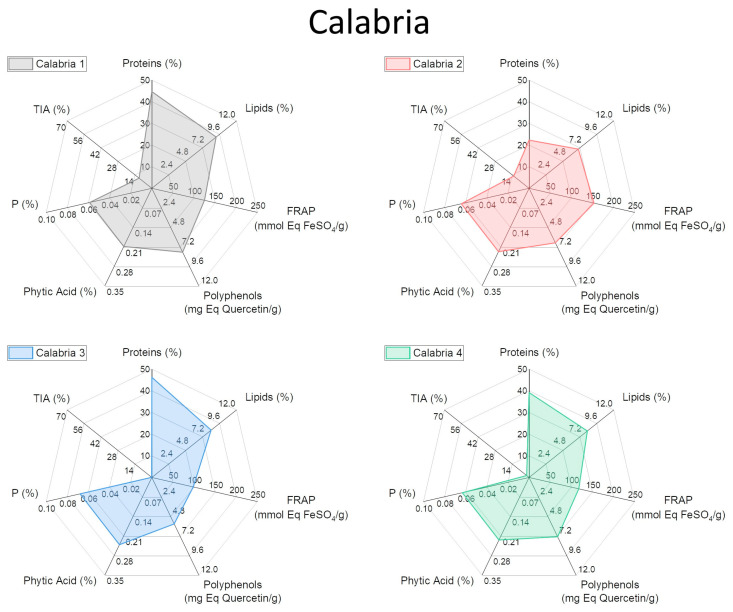
Radar plot representation of the main biochemical characteristics of white lupin landraces collected in the Calabria region. See Appendix A for detailed information.

**Figure 6 plants-13-00785-f006:**
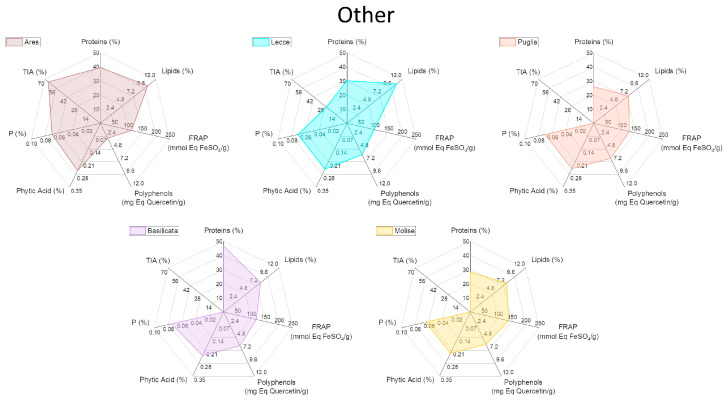
Radar plot representation of the main biochemical characteristics of commercial reference Ares lupin and white lupin landraces collected in the Molise, Puglia and Basilicata regions. See Appendix A for detailed information.

**Figure 7 plants-13-00785-f007:**
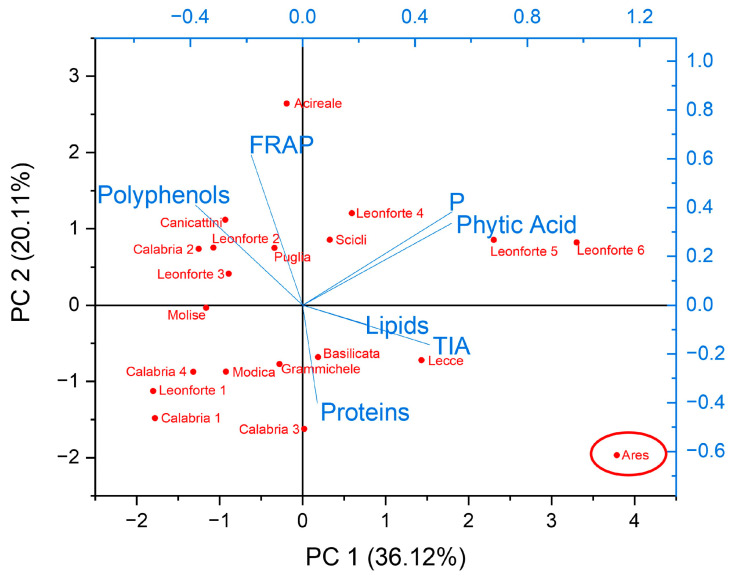
Principal Component Analysis. Biplot of Principal Component 1 and Principal Component 2. In blue is reported the loading plot with the eigenvectors of the variables, and in red is reported the score plot with the distribution of landraces. Multivariate analysis was performed on the data reported in Appendix A. Commercial reference is highlighted with a red circle.

**Figure 8 plants-13-00785-f008:**
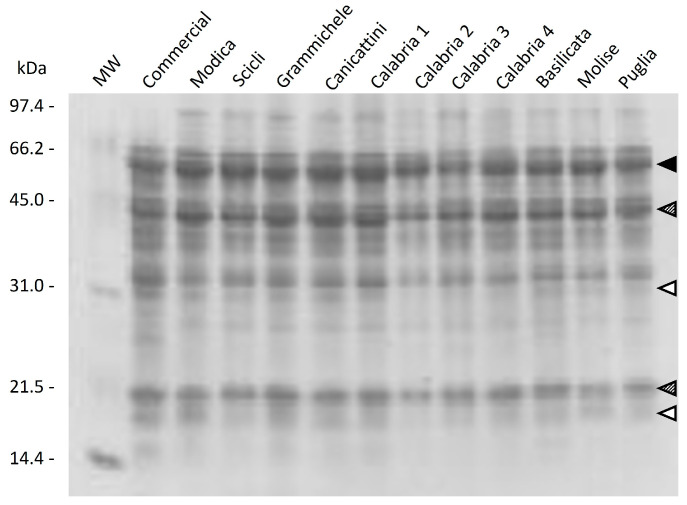
Qualitative protein characterization. Total extractable seed proteins stained with CBB. Shaded arrowheads: α-conglutin major and minor subunits; black arrowhead: β-conglutin polypeptides; white arrowheads: γ-conglutin major and minor subunits. MW: Molecular weight standard markers.

**Table 1 plants-13-00785-t001:** List of investigated white lupin genotypes. Nd: not determined.

Name	Type	Region/Province of Collection
Leonforte 1	Landrace	Enna (Sicily, Italy)
Leonforte 2	Landrace	Enna (Sicily, Italy)
Leonforte 3	Landrace	Enna (Sicily, Italy)
Leonforte 4	Landrace	Enna (Sicily, Italy)
Leonforte 5	Landrace	Enna (Sicily, Italy)
Leonforte 6	Ecotype	Enna (Sicily, Italy)
Acireale	Landrace	Catania (Italy)
Canicattini	Landrace	Siracusa (Sicily, Italy)
Modica	Landrace	Ragusa (Sicily, Italy)
Scicli	Landrace	Ragusa (Sicily, Italy)
Grammichele	Landrace	Catania (Sicily, Italy)
Calabria 1	Landrace	Calabria (Italy)
Calabria 2	Landrace	Calabria (Italy)
Calabria 3	Landrace	Calabria (Italy)
Calabria 4	Landrace	Calabria (Italy)
Puglia	Landrace	Apulia (Italy)
Lecce	Landrace	Apulia (Italy)
Basilicata	Landrace	Basilicata (Italy)
Molise	Landrace	Molise (Italy)
Ares	Commercial	Nd

**Table 2 plants-13-00785-t002:** Plants phenological stages (days to sowing time). Different letters in the same column indicate significant differences (*p* ≤ 0.001) based on Tukey’s HSD.

Sample	End Germination–Beginning Plant Emergence	Main Stem Flowering	Main Pods Growth	Pod and Seed Maturity
Leonforte 1	12.0 ± 0.00 bc	93.0 ± 0.00 gh	106.5 ± 0.71 ghi	175.0 ± 0.00 def
Leonforte 2	11.5 ± 0.71 bc	89.5 ± 0.71 hi	103.0 ± 1.41 hi	174.5 ± 0.71 ef
Leonforte 3	13.0 ± 0.00 abc	103.5 ± 0.71 e	118.5 ± 0.71 cdef	176.0 ± 0.00 cdef
Leonforte 4	12.5 ± 0.71 abc	96.0 ± 0.00 g	111.0 ± 0.00 g	175.0 ± 0.00 def
Leonforte 5	12.0 ± 0.00 bc	91.5 ± 0.71 hi	107.5 ± 0.71 gh	175.0 ± 0.00 def
Leonforte 6	11.0 ± 0.00 c	80.5 ± 0.71 jk	95.5 ± 0.71 j	174.0 ± 0.00 f
Acireale	11.0 ± 0.00 c	81.5 ± 0.71 j	97.5 ± 0.71 j	174.0 ± 0.00 f
Canicattini	11.0 ± 0.00 c	80.0 ± 0.00 jk	95.0 ± 0.00 j	174.0 ± 0.00 f
Modica	12.0 ± 0.00 bc	93.5 ± 0.71 gh	106.5 ± 0.71 ghi	175.0 ± 0.00 def
Scicli	13.0 ± 0.00 abc	104.5 ± 0.71 de	117.5 ± 0.71 ef	176.5 ± 0.71 bcde
Grammichele	11.0 ± 0.00 c	77.5 ± 0.71 k	90.5 ± 0.71 k	174.0 ± 0.00 f
Calabria 1	13.0 ± 0.00 abc	106.5 ± 0.71 cde	119.5 ± 0.71 bcde	176.5 ± 0.71 bcde
Calabria 2	13.0 ± 0.00 abc	104.0 ± 0.71 e	118.0 ± 0.71 def	175.5 ± 0.71 def
Calabria 3	15.0 ± 0.00 a	111.5 ± 0.71 a	121.5 ± 0.71 abcd	180.0 ± 0.00 a
Calabria 4	13.0 ± 0.00 abc	107.5 ± 0.71 cd	122.5 ± 0.71 ab	177.0 ± 0.00 bcd
Puglia	14.0 ± 1.41 ab	108.0 ± 0.00 b	123.0 ± 0.00 ab	178.0 ± 0.00 abc
Lecce	13.0 ± 0.00 abc	106.0 ± 0.00 cde	122.0 ± 0.00 abc	176.0 ± 0.00 cdef
Basilicata	12.5 ± 0.71 abc	99.5 ± 0.71 f	115.5 ± 0.71 f	175.0 ± 0.00 def
Molise	15.0 ± 0.00 a	109.0 ± 0.00 ab	124.0 ± 0.00 a	178.5 ± 0.71 ab

## Data Availability

Data are contained within the article and Appendix A.

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
