# Peer review of "Biochemical Characterization of the Seed Quality of a Collection of White Lupin Landraces from Southern Italy"

_plants, 2024, doi:10.3390/plants13060785_

Round 1
Reviewer 1 Report
Comments and Suggestions for Authors
In my opinion, the manuscript entitled Biochemical characterization of the seed quality of a collection of white lupin landraces from Southern Italy by Spina et al., aimed to realize a characterization of 19 L. albusseeds. Firstly, authors realized a simple description of the samples and afterward they characterize their proximate composition: protein, fat but also total polyphenols, phytic acid, TIA and molecular fractions through SDS PAGE analysis. The methods are quite enough described, and the results are discussed and compared with the current state of the art. I have some small comments, as follows:
1. Line 62. Please mentioned the period of time (years), instead of some years. It is not an academic expression.
2. At table 2 please only mentioned the title of the table. Results ……..till lines 139 please removed under the table.
3. Line 271. After staining with Coomassie Blue which solution was used for the distaining of the gels? What was the time needed for the complete running of the gels? which equipment was used for SDS PAGE?
4. Line 338 – please mentioned the equation used for the calibration curve and calculation of the total phenols content?
5. About my opinion, it is quite strange to not have some differences in the molecular protein fraction through SDS PAGE. This is a sensitive analyze and the intensity of the bands are quite different mainly on Caniccattini, Grammichele and Calabria 1, at a molecular fraction of 45 kDa. Did authors calculate the protein coefficient degradation? Please better discussed this part of the SDS page analysis and better justified the results.

Comments on the Quality of English LanguageMinor editing of English language required
Author Response
Q: Line 62. Please mentioned the period of time (years), instead of some years. It is not an academic expression.
ANSWER We modified “some years” with “a decade” in the text, now at line 69 adding a reference [24] Romeo, F.V.; Fabroni, S.; Ballistreri, G.; Muccilli, S.; Spina, A.; Rapisarda, P. Characterization and Antimicro-bial Activity of Alkaloid Extracts from Seeds of Different Genotypes of Lupinus Spp. Sustainability 2018, Vol. 10, Page 788 2018, 10, 788, doi:10.3390/SU10030788.
Q: At table 2 please only mentioned the title of the table. Results ……..till lines 139 please removed under the table.
ANSWER The sentence: “Significance level was set at 0.05 and it is reported in Table 2 with the lowercase letters in the brackets, the same letter means the difference is not statistically significant.” Was moved to the material and methods – statistical analysis section. “Mean comparison was performed with Tukey’s Test” was removed from table 2 caption since already specified in the material and methods – statistical analysis section. The explanation of abbreviations was left in the table caption as requested by reviewer 2
Q: Line 271. After staining with Coomassie Blue which solution was used for the distaining of the gels? What was the time needed for the complete running of the gels? which equipment was used for SDS PAGE?
ANSWER We added in the material and methods the time for the gel run (90 min), the SDS-PAGE apparatus (Mini-Protean III Cell (Bio-Rad Laboratories, Inc., Hercules, CA, USA) and the composition of the destaining solution (10% ethanol, 10% acetic acid solution). Now at line 294.
Q: Line 338 – please mentioned the equation used for the calibration curve and calculation of the total phenols content?
ANSWER More details were added as requested in the description of the calibration curve for determination of total phenolic content: “A calibration curve was prepared with serial dilutions of a quercetin standard (Sigma-Aldrich, St. Louis, MO, USA) concentrated 0.5 mg/L, in a range comprised from 0.5 mg/L and 0.025 mg/L. The linear equation obtained from the standard curve was used to interpolate each sample total phenolic extract concentration and the results were reported as milligrams of quercetin equivalents (QE) per gram of flour.” Now at line 364.
Q: About my opinion, it is quite strange to not have some differences in the molecular protein fraction through SDS PAGE. This is a sensitive analyze and the intensity of the bands are quite different mainly on Caniccattini, Grammichele and Calabria 1, at a molecular fraction of 45 kDa. Did authors calculate the protein coefficient degradation? Please better discussed this part of the SDS page analysis and better justified the results.
ANSWER As requested SDS-PAGE analysis were discussed to highlight similarities and differences among the samples. The following paragraph “The band corresponding to the main seed proteins, namely a-, β- and γ-conglutin, are evident in all the analyzed samples, regardless of their total protein content as indicated by arrowheads in Figure 7. Overall, the electrophoretic approach showed that the protein profile of the analyzed landraces was very similar among each other and as well to the commercial reference Ares. However, some differences can be highlighted in the intensity of specific band on the gel. Compared to the control, the landraces have a lower relative content in γ-conglutin, especially Calabria 1 and Calabria 2. This result highlights that commercial Ares is rich in the hypoglycemic protein. Instead, the amount of bconglutin, the protein involved in plant defense, is similar in all samples analyzed.” The paragraph was added at line 236
Reviewer 2 Report
Comments and Suggestions for Authors
Dear Authors,
The scientific value of the paper in its current form has increased compared to the previous version of the manuscript. I still have some comments:
1) Explanations of abbreviations used in the tables should be included below the tables.
2) The Authors decided to present the same data again, not only in a table 2, but also graphically, using a radar chart (Figures 2-5). I stand by my opinion that presenting the same data twice, in table 2 and in radar charts, is not appropriate.
3) L 90-91: The Authors state "Results are reported as grams of protein per 100g of milled seeds"- at what moisture content? Were the contents of the determined components converted to dry weight ? I suggest giving the nutrient content not in % but in mass units per kg of dry matter.
Author Response
1) Explanations of abbreviations used in the tables should be included below the tables.
ANSWER: The explanation of the abbreviation was added in table 2 caption: ”FRAP: Ferric Reducing Ability of Plasma; TIA: Trypsin inhibitory activity”
2) The Authors decided to present the same data again, not only in a table 2, but also graphically, using a radar chart (Figures 2-5). I stand by my opinion that presenting the same data twice, in table 2 and in radar charts, is not appropriate.
ANSWER: We understand reviewer’s point since the data reported in the table and in the graphs are of course the same, however, we think that removing one of the two representations will weaken the dissertation. The numerical report in table 2 allows the accurate quantification and statistical comparison of the data, while the graphical report helps the simultaneous qualitative (and semi-quantitative) comparison of the grain’s characteristics, supporting the discussion aimed to describe the landraces. On the contrary, the graph representation although facilitating the direct comparison, lacks of the strictness of the numerical data and of the statistical significance. For these reasons, for a more complete depiction, we decided to keep both the numerical table and the graphical visualization of the data.
3) L 90-91: The Authors state "Results are reported as grams of protein per 100g of milled seeds"- at what moisture content? Were the contents of the determined components converted to dry weight ? I suggest giving the nutrient content not in % but in mass units per kg of dry matter.
ANSWER: We thank the reviewer for raising this point, since we noticed that we were inaccurate in reporting the methods. Seeds, indeed, have been drier prior to any analysis, thus the reported results already refer to the dry weight. We added a sentence in 3. Materials and Methods 3.1. Plant Material and Sampling to clarify this point: “Seeds were dried at 50°C in an oven until no further weight loss was recorded (48 h), finely ground to 60 mesh particles and stored in sealed jars.” at line 273. We edited accordingly also the table 2 caption. In table 2 as well as in the text the nutrient content characterizations (Protein, Lipid, Phytic acid and Phosphorous) were expressed as g/kg dry weight.
Reviewer 3 Report
Comments and Suggestions for Authors
This manuscript much likes an experimental report, lacking discussion part (althoug you list the "results and discussion" part). Please add some to highlight the characteristics and values of these lupin resources. By the way, please provide the protein extracts amount of each sample for SDS-PAGE.
---
Additional comments :
line 73-87, they belongs to the "Materials and Methods" part line 89-91, please put them in the "Materials and Methods" part line 94-96, it is not accurate. line 103, what is common value? line 104-106, nonesense line 112-120, what can you get from them? line 124-128, you have present the phosphorous content, why not describe them directly? line 140, Why is the TIA so high in the reference variety of Ares and landraces low? On the one hand, you say that this is antinutrientional factor, why is it so high in the commercial variety, it seems illogical. And why there is no data of hundred seeds weight for Ares, and it is a simple trait and easy to get. line 141-186, the radar plots are repeat of Table 2, is it necessary? line 212-221, it is better to put them in "introduction" part. line 226-228, please describe the results firstly. line 238, please add the introduction of the reference variety of Ares in this part. Comments on the Quality of English LanguageEnglish is ok.
Author Response
QUESTION: This manuscript much likes an experimental report, lacking discussion part (althoug you list the "results and discussion" part). Please add some to highlight the characteristics and values of these lupin resources. By the way, please provide the protein extracts amount of each sample for SDS-PAGE.
ANSWER: We expanded and got into detail with the discussion section as requested adding the following two texts:
- “The band corresponding to the main seed proteins, namely a-. β- and γ-conglutin, are evident in all the analyzed samples, regardless of their total protein content as indicated by arrowheads in Figure 7. Overall, the electrophoretic approach showed thatthe protein profile of the analyzed landraces was very similar among each other and as well to the commercial reference Ares. However, some differences can be highlighted in the intensity of specific band on the gel. Compared to the control, the landraces have a lower relative content in γ-conglutin, especially Calabria 1 and Calabria 2. This result highlights that commercial Ares is rich in the hypoglycemic protein. Instead, the amount of -conglutin, the protein involved in plant defense, is similar in all samples analyzed. Now at line 236.
“Finally, the selected landraces can be clustered in two different subgroups i.e. those with high protein content (lower left panel of Bi-plot) and those with high polyphenol content (upper left panel of the Bi-plot). The attribution of peculiar nutrient content profiles to each landrace may help to select the appropriate one as a function of the desired application. Lupin indeed, due to its high protein content, is a good gluten-free meat substitute in vegetarian and vegan diets, furthermore, due to its low lipid content it could be even better than soybean whose, nowadays, is the gold standard. From a technological point of view instead a high content of polyphenols and/or molecules with a reducing power could help in extending shelf life or enhance food properties in particular products, adding to the other well-known emulsifying, foaming and gelling properties of lupin seeds [35].” Adding also the following reference Duranti, M.; Gius, C. Field Crops Research Legume Seeds: Protein Content and Nutritional Value. Field Crops Res 1997, 53, 31–45. (lines 219-228)
We added in the Material and method section 3.3 SDS-PAGE the sentence: “15 µL per sample were loaded on the gel.” (line 292)
QUESTION: line 73-87, they belongs to the "Materials and Methods" part.
ANSWER: Most of the information was moved in the “Material and Methods” part as requested. However, we left some key information to frame the samples used in the experiments. A description of the landraces and the visualization of their origin in the text is necessary to better understand the focus of the work based on the different origins of samples.
QUESTION: line 89-91, please put them in the "Materials and Methods" part. ANSWER: Lines was moved in the “Material and Methods” part as requested.
QUESTION: line 94-96, it is not accurate.
ANSWER: We changed the term “ comparable” with “not statistically different from”
QUESTION: line 103, what is common value?
ANSWER: We changed the term “commonly” with “what can be found in literature”
QUESTION: line 104-106, nonesense
ANSWER: We specified soybean oil content with a new reference (Medic, J.; Atkinson, C.; Hurburgh, C.R. Current Knowledge in Soybean Composition. J Am Oil Chem Soc 2014, 91, 363–384, doi:10.1007/s11746-013-2407-9.), to highlight that lupin is not interesting for its oil content, if compared with other pulses such as soybean. The text was edited with “. Because of this low lipid content, lupin is not interesting for oil extraction, unlike soybean whose oil content is up to 24%” at line 109.
QUESTION: line 112-120, what can you get from them?
ANSWER: We added the following sentence to comment the comparison of our landraces to others reported in literature where polyphenol content was evaluated: “The landraces selected for this study, thus, have a polyphenol content similar to what reported in literature.” at line 126.
QUESTION: line 124-128, you have present the phosphorous content, why not describe them directly?
ANSWER: We added the following sentence: “Phosphorous content is constant overall the analyzed landraces as well as Phytic acid content.” at line 133.
QUESTION: line 140, Why is the TIA so high in the reference variety of Ares and landraces low? On the one hand, you say that this is antinutrientional factor, why is it so high in the commercial variety, it seems illogical. And why there is no data of hundred seeds weight for Ares, and it is a simple trait and easy to get.
ANSWER: In our opinion the data it’s not so strange since the commercial variety are selected to improve positive characteristics and reduce the negative ones for the consumer such as the bitter taste. TIA is an antinutritional factors but it isn’t a risk since, for human consumption, the seed isn’t eaten raw and trypsin inhibitors are inactivated by heat treatment since lupin seeds are eaten cooked. We added 100 seed weight of Ares.
QUESTION: line 141-186, the radar plots are repeat of Table 2, is it necessary? ANSWER: We understand reviewer’s point since the data reported in the table and in the graphs are of course the same, however, we think that removing one of the two representations will weaken the dissertation. The numerical report in table 2 allows the accurate quantification and statistical comparison of the data, while the graphical report helps the simultaneous qualitative (and semi-quantitative) comparison of the grain’s characteristics, supporting the discussion aimed to describe the landraces. On the contrary, the graph representation although facilitating the direct comparison, lacks of the strictness of the numerical data and of the statistical significance. For these reasons, for a more complete depiction, we decided to keep both the numerical table and the graphical visualization of the data.
QUESTION: line 212-221, it is better to put them in "introduction" part.
ANSWER: The lines were inserted in “Introduction” and the text was modified accordingly.
QUESTION: line 226-228, please describe the results firstly.
ANSWER: The text was modified and the description of results implemented.
QUESTION: line 238, please add the introduction of the reference variety of Ares in this part.
ANSWER: We moved the introduction of the Ares reference material at the end of the paragraph: “Furthermore, a commercial lupin breeding (Lupinus albus Ares) was used as reference material.” Now at line 273.
Round 2
Reviewer 1 Report
Comments and Suggestions for Authors
In my opinion, authors have highly improved the scientific quality of the paper, so, it can be published.
Thank you!
Comments on the Quality of English LanguageMinor editing of English language required
Author Response
Thanks.
Reviewer 3 Report
Comments and Suggestions for Authors
You have made responses and modifications according to my concerns , and this is very good. However, I think you also need detailed proofreading to ensure that the work is written scientifically, professionally, and accurately. I've listed some of the questions below, but you have to check the whole manuscript.
line 105-110, The protein content is significantly lower by an order of magnitude, and you'd better check all the data in detail, including the contents of other ingredients.
line 212-215, Please add significant signs for the correlation coefficients on Figure S1.
line 228, Figure 6, no confidence area, no arrows. I don't think you fully understand PCA analysis, let alone you derive classifications based on proteins and polyphenols.
line254-258, What are the criteria of the highest and lowest protein contents?
line 273, Figure 7, I don't think the picture is clear enough, including the protein ladder, so it's hard to draw any conclusions. Also, because of the problem with your classification, I don't see you making a direct comparison of the protein compositions within or between these different groups of lupin varieties.
line 307, pH 9.0?
Comments on the Quality of English LanguagePlease see above.
Author Response
QUESTION 1: You have made responses and modifications according to my concerns , and this is very good. However, I think you also need detailed proofreading to ensure that the work is written scientifically, professionally, and accurately. I've listed some of the questions below, but you have to check the whole manuscript.
ANSWER 1: Thanks for the comment. The whole manuscript has been carefully checked.
QUESTION 2: line 105-110, The protein content is significantly lower by an order of magnitude, and you'd better check all the data in detail, including the contents of other ingredients.
ANSWER 2: Protein, lipid, phosphorous and phytic acid content of lupin landraces is reported in our paper as g/kg of dry matter (as requested by another Reviewer). 1% = 10 g/kg. Other papers report their content as percentage. To facilitate comparison of the data we reported percentages in brackets. Thus, just for the sake of clarity, the protein content of Calabria 2 landrace (lines 96-97) is 223 g/kg or 22.3 %.
QUESTION 3: line 212-215, Please add significant signs for the correlation coefficients on Figure S1.
ANSWER 3: We might have misunderstood the reviewer’s comment, but we weren’t able to find literature supporting the necessity of adding significant signs to a correlation coefficient. The correlation coefficient is a measure of the association between two variables. The correlation coefficient can be interpreted as the scalar product of two standardized variables. Given that the distance to the origin of these variables is equal to one, we see that the correlation between two variables coincides with the cosine of the angle formed by them. The correlation coefficient ranges between -1 and +1. When the correlation is equal to 1, this indicates that there is a perfect linear association between the two variables. A correlation coefficient equal to -1 indicates that there is a perfect inverse linear association.
QUESTION 4: line 228, Figure 6, no confidence area, no arrows. I don't think you fully understand PCA analysis, let alone you derive classifications based on proteins and polyphenols.
ANSWER 4: PCA loading plot is commonly reported with a simple line without arrowheads, indeed the plot presented in the paper was exported as such from the software used for statistical analysis, OriginPro 2023 (as stated in M&M), one of the most powerful and commonly used software for data analysis. Confidence area was not reported since it included all the samples analyzed except for Ares. Thus, failing to discriminate different populations among the samples reported, hardly possible since dealing with landraces of the same species, we still discussed the most relevant variables (i.e. protein and polyphenol content) characterizing the samples, as evidenced by the loading plot vectors. Anyway, we decided to remove from the discussion the following sentence: “Finally, the selected landraces can be clustered in two different subgroups i.e. those with high protein content (lower left panel of Bi-plot) and those with high polyphenol content (upper left panel of the Bi-plot).”
QUESTION 5: line254-258, What are the criteria of the highest and lowest protein contents?
ANSWER 5: We selected the landraces with the highest and the lowest protein content among our sample set, to verify if there were differences related to the quality of protein content.
QUESTION 6: line 273, Figure 7, I don't think the picture is clear enough, including the protein ladder, so it's hard to draw any conclusions. Also, because of the problem with your classification, I don't see you making a direct comparison of the protein compositions within or between these different groups of lupin varieties.
ANSWER 6: We agree with the Reviewer that the molecular weight marker is barely visible; however, it is enough visible to attribute the correct molecular weights to the ladder. Furthermore, the protein pattern and the main seed proteins, namely α-, β-, and γ-conglutin, are clearly evident and comparable to other SDS-PAGE reported in the literature. For sake of clarity, we indicated two references (line 251)
We therefore edited the sentence at lines 257-258 to mitigate the claim of the discussion. It now is: “This result highlights that commercial Ares is slightly richer in the hypoglycemic protein” (line 258)
QUESTION 7: line 307, pH 9.0?
ANSWER 7: 9.0 is the pH value of the extraction buffer. This pH is in the range of the buffering capacity of TRIS-HCl system. We removed the comma and slightly changed the sentence to help the comprehension. Now sentence is now: “One gram of each flour was resuspended in 20 mL (ratio 1:20 w/v) of 50 mM Tris-HCl buffered at pH 9.0, containing 300 mM NaCl.”